# Bioactivity and Biocompatibility Properties of Sustainable Wollastonite Bioceramics from Rice Husk Ash/Rice Straw Ash: A Review

**DOI:** 10.3390/ma14185193

**Published:** 2021-09-10

**Authors:** Hamisah Ismail, Hasmaliza Mohamad

**Affiliations:** School of Materials and Mineral Resources Engineering, Universiti Sains Malaysia, Nibong Tebal 14300, Penang, Malaysia; hamisahismail@usm.my

**Keywords:** bioactivity, biocompatibility, wollastonite, rice husk ash, rice straw ash

## Abstract

Recently, there has been an increase in interest in agricultural waste in scientific, technological, environmental, economic, and social contexts. The processing of rice husk ash/rice straw ash into biocompatible products—also known as biomaterials—used in biomedical implants is a technique that can enhance the value of agricultural waste. This method has effectively converted unprocessed agricultural waste into high-value products. Rice husk and straw are considered to be unwanted agricultural waste and are largely discarded because they pollute the environment. Because of the related components present in bone and teeth, this waste can produce wollastonite. Wollastonite is an excellent material for bone healing and implants, as well as tissue regeneration. The use of rice husk ash or rice straw ash in wollastonite production reduces the impact of agricultural waste on pollution and prompts the ensuing conversion of waste into a highly beneficial invention. The use of this agricultural waste in the fabrication of wollastonite using rice husk ash or rice straw ash was investigated in this paper. Wollastonite made from rice husk ash and rice straw ash has a fair chance of lowering the cost of bone and tooth repair and replacement, while having no environmental effects.

## 1. Introduction

Agricultural waste is any waste that was produced in accumulative concentrations because of various farming processes [1]. The effective use of agricultural waste mitigates environmental issues caused by reckless waste disposal. Agricultural waste management is a critical approach to environmental waste management. If some waste is abundant, it can significantly affect the ecosystem for humans, livestock, and plants [2]. The nature, amount, and types of waste produced in agriculture vary across countries. The quest for an efficient way to handle agricultural waste properly would enhance health quality and protect the environment.

Waste can be reused, recycled, and guided into the manufacture of value-added goods for sustainable growth. This protects the environment on the one hand, and on the other creates a zero-waste standard to obtain value-added goods. The use of waste today is a priority for the achievement of sustainable development [3].

Rice husk or rice straw was used as a precursor for biomaterial, biomedical implants, and surgery applications, because agricultural waste of this type adds considerable value. Using agricultural waste to create value-added products has added a new dimension to producing biocompatible materials or biomaterials from agricultural waste. This is feasible because some of the waste contains active compounds that are needed for biomedical purposes. This is a modern medical procedure that is expected to be necessary. Most agricultural waste studies have focused primarily on its energy potential or use as functional renewable raw materials and an efficient chemical feedstock in producing valuable products, due to its abundance, low cost of availability, and renewability [4,5,6,7,8,9,10,11]. Microorganisms or their components carry out this conversion into valuable goods or energy sources [12,13]. The use of biomass as a renewable energy source, or for the production of biogenic materials—especially from rice husk and rice straw—has sparked a lot of interest in recent years [14,15,16]. Previous studies reported that agricultural waste is a safe substitute used as bioceramics or bioglass [17,18,19], biomaterials in bone substitution to stimulate osteoblast development [10] in medical and dental therapy [20,21,22,23,24,25], and anticancer drug delivery [26].

Because of the increased number of patients needing bone replacement—specifically those with bone cancer, trauma, or advanced age—biomaterials’ innovation for bone tissue replacement has increased and piqued many researchers’ interest. Before being used as bone implants, biomedical devices must be biocompatible, with adequate mechanical strength to sustain human body weight [26,27,28].

This review concisely highlights the recent research results on the utilization of rice husk (RH), rice husk ash (RHA), rice straw (RS), and rice straw ash (RSA)—agricultural waste products—to produce wollastonite bioceramics, for the first time, and Figure 1 overviews briefly the core outline of this review. It was expected that this review could perk the successful development of rice husk and rice straw, which broadly remain not fully utilized and untransformed.

## 2. Biomaterials

Biomaterials are natural or synthetic materials used in medical applications to perform bodily functions or replace damaged body parts or tissues [29]. The initial definition of biomaterials is any inert materials used in medical devices to interact with the biological system [30]. They can be used as a medium for drugs or drug delivery systems. Biomaterials can generally be classified as biological and synthetic materials, as presented in Figure 2. Examples of biological biomaterials are collagen and chitin. Synthetic biomaterials consist of four main types: ceramics, metals, polymers, and composites. Examples of medical ceramics include phosphate ceramics, ceramics, and glass-ceramics. Ceramics and glass-ceramics are widely used in the fields of dentistry and bone replacement. Table 1 compares the structure and structural limits of adult human calcified tissues’ inorganic phases.

### 2.1. Bioceramics

Since the 1960s, a revolution has taken place in the consumption of ceramics as an implant material to improve human health quality. This revolution involved developing the production and fabrication of ceramics as implants explicitly designed to repair and reconstruct bones in response to disease, damage, or wear on limbs. Referring to Hench [33], the ceramics used for this aim are known as bioceramics. Most bioceramic materials are related to the repair of skeletal systems—consisting of bones, joints, teeth, and an intermediate medium for hard and soft tissues. Ceramics are also applied to substitute components of the cardiovascular system, particularly heart valves.

Ceramics are usually inorganic and non-metallic materials, which are chemically stable and have excellent thermal resistance properties. The use of ceramics is widely applied in the reconstruction of hard tissue rather than soft tissue, because they have excellent strength, wear resistance, and high durability. Moreover, ceramics also have many advantages, including good biochemical properties, hemocompatibility, non-toxicity, ease of production and molding, sterility, and non-immunogenicity. Bioceramics can be categorized into three generations [34], namely:First generation: Inert bioceramics

Alumina, titanium, and zirconia are commonly used in dental work, orthopedics, and implants [34,35,36,37]. These substances do not significantly affect body tissues when implanted, as they are very inert chemically or biologically.

Second generation: Bioactive and bioresorbable bioceramics

Calcium silicate, or wollastonite (CaSiO_3_), is a bioceramic material used in the biomedical field based on its bioactive properties, which are potentially used in bone recovery [38,39]. It is easily biodegradable, and its insoluble substances (Ca and Si) help bone cells’ development and function. Bioactive Ca–Si materials are classified as bioactive ceramics, bioactive glasses, glass-ceramics, and cements.

On the other hand, bioresorbable refers to a substance that dissolves slowly, changing according to the state of the tissue in the human body, when inserted into the human body. Examples of bioresorbable materials include calcium carbonate (CaCO_3_), tricalcium phosphate (TCP, Ca_3_(PO_4_)_2_), calcium oxide (CaO), and polyactic-polyglycolic copolymeric acid, among other examples of bioresorbable materials that are commonly used today [40,41,42]. TCP is widely used as bioresorbable material in hard tissue replacement, which provides a place for new bone tissue to grow and dissolve in the bone marrow [43,44], before leaving the new bone after complete reabsorption.

Third generation: Driving the living tissues generation

The third generation of biomaterials applies bioactive and bioresorbable materials as temporary three-dimensional porous structures or scaffolds capable of stimulating genes that induce living tissue regeneration. Concepts of bioactivity and biodegradability are mixed with these biomaterials, and this convergence of both concepts tends to be a crucial aspect for third-generation biomaterials. Therefore, the main purpose of third-generation bioceramics is to provide an appropriate scaffolding structure that enables living cells to perform their natural processes. Furthermore, the principle of porosity and its range of order tend to be of primary importance. Bioceramics with mesoporosity between 2 and 50 nm are used for drug and biologically active molecule applications, where they are loaded and then released to help in the bone substitution process. As scaffolds for tissue engineering, macroporous materials with pore dimensions exceeding many microns tend to be ideal. Examples include mesoporous silica [45], ordered mesoporous glasses [46], porous scaffolds of calcium phosphate [47,48], and organic/inorganic hybrids [49,50] with cellular knowing positions.

To finalize this section, it should be emphasized that the first generation of inert bioceramics was intended to serve as artificial bone grafts. The second generation of bioactive and bioresorbable bioceramics was designed to mimic some biomineralization-related functions. In contrast, the third generation of bioceramics effectively provides an adequate scaffolding structure supporting bone cell functions. Figure 3 shows a summary of the three generations of bioceramics.

### 2.2. Wollastonite

Wollastonite is a mineral from the silicate group and was named in honor of Sir William Hyde Wollaston (1766–1828). It consists of the elements calcium (Ca), silicon (Si), and oxygen (O), with the chemical formula CaSiO_3_. Theoretically, wollastonite’s composition consists of 48.38% calcium oxide (CaO) and 51.78% silicon dioxide (SiO_2_). Although most wollastonite is relatively pure, it can consist of other elements, such as aluminum, iron, manganese, magnesium, potassium, sodium, or the element calcium with strontium in its mineral structure. The color of natural wollastonite is white, and changes color to grey-white, yellow, light green, pinkish, brown, or red in the presence of other elements or impurities [51]. Table 2 generally displays the physical and chemical properties of wollastonite.

Figure 4 depicts the phase diagram of the calcium oxide—silicon dioxide (CaO–SiO_2_) method, demonstrating the relationship between temperature and the percentage weight of CaO and SiO_2_ producing the wollastonite phase. From Figure 4, it can be seen that two types of wollastonite could form with temperature change, even though the composition of CaO–SiO_2_ is still the same. In other words, wollastonite has polymorphic and polytype forms. In general, wollastonite has two polymorphic forms—specifically, pseudo-wollastonite (α-CaSiO_3_) and beta-wollastonite (β-CaSiO_3_) [52]. α-CaSiO_3_ is formed at high temperatures exceeding 1125–1436 °C. while β-CaSiO_3_ is formed at low temperatures not exceeding 1125 °C.

#### 2.2.1. Low-Temperature Wollastonite (β-CaSiO_3_)

Low-temperature wollastonite (β-CaSiO_3_) is widely found in calcium-rich rocks undergoing metamorphic processes and areas undergoing regional metamorphism at low pressures, and is often associated with calcite, grossular, vesuvianite, akermanite, larnite, and spurit-β-CaSiO_3_ shows triclinic structure [53].

#### 2.2.2. High-Temperature Wollastonite or Pseudo-Wollastonite (α-CaSiO_3_)

High-temperature wollastonite (α-CaSiO_3_) is also known as pseudo-wollastonite. Natural pseudo-wollastonite is a newly discovered constituent of ultra-high-combustion metamorphism and igneous rocks, found in calcium-rich, rock-forming minerals found in volcanic mud fossils (found in the Dead Sea area) [54]. Pseudo-wollastonite is the result of combustion metamorphism caused by ignition of hydrocarbons. The melt begins to crystallize in the temperature range of 1480–1500 °C, without any pressure. The pseudo-wollastonite consists of two groups of minerals: the first contains rankinite, larnite, nagelschmidtite, wollastonite (1T) gehlenite-melilite mixture, andradite-Ti mixture, cuspidine, and fluorapatite; while the second group of minerals includes parawollastonite (2M), wollastonite (1T), melilite-gehlenite mixture, and fluorellestadite [54].

#### 2.2.3. Wollastonite in the Biomedical Field

Commercial bioactive glass-ceramics used as implants in the medical industry include CERABONE^®^, made from apatite–wollastonite glass-ceramics, CERAVITAL^®^, made from apatite–devitrite glass-ceramics, and BIOVERIT^®^ I, from mica–apatite glass-ceramics [55]. Bioactive glass, such as BIOGLASS^®^, is used on the head and in throat surgery’s middle ear device. CERABONE^®^ AW, manufactured by Nippon Electric Glass Co., Ltd., Shiga, Japan, is a bioactive glass made from apatite–wollastonite glass-ceramics. It is most widely used as a bone replacement in human medicine. CERABONE^®^ glass-ceramics show excellent bioactive properties. The mechanism of apatite formation reaction between CERABONE^®^ with living bone (animal bone) and SBF solution without the presence of living bones it is still able to form apatite with the release of Ca^2+^ ions, and the reaction of the Si–OH group on CERABONE glass-ceramics’ surface allows the apatite formation reaction to take place, while SBF provides sufficient phosphate ions for the formation of apatite on the surface of the CERABONE.

## 3. Description of Agricultural Waste

Agriculture residues are further classified as process residues and field residues. Field residues are residues present in the field as a result of the harvesting process, including stalks, leaves, stems, and seed pods; while process residues remain residues even after the crop has been converted into a beneficial alternative raw material, and include husks, leaves, stalks, straw, bagasse, seeds, stems, molasses, shells, peel, stubble, pulp, and roots used in animal feed, fertilizer manufacturing, soil improvement, and a variety of other processes. According to a 2012 National Solid Waste Management (JPSPN) study, approximately 31–45% of the total volume of waste generated each day is from agricultural—amongst the highest generated waste produced in Malaysia [56]. The household sector produces about 44.5%, and the ICI (institutional, commercial, and industrial) sector produces about 31.4% of this waste [56]. With rising production in the region, the proportion of the waste generated by the country also increased. Most of this waste is left wasted or untreated, having harmful effects on climate, human health, and animal health. However, its composition includes a significant quantity of organic compounds that have created various value-added goods and lowered manufacturing costs, as illustrated in Table 3.

### 3.1. Paddy

Paddy is generally a semi-aquatic seasonal crop, which can live either in very fluid reservoirs or in hilly areas. Paddy, or its scientific name *Oryza sativa*, is the main crop in the northern part of the peninsula compared to other areas in Malaysia, while huma paddy, or hill paddy, is only grown on small-scale highlands. Rice is the world’s second most popular food, with rice paddy production reaching about 700 million tonnes in 2019 [14,64]—a figure that is expected to rise gradually as the world’s population grows.

#### 3.1.1. Rice Husk and Rice Husk Ash

Rice husk is a byproduct of the milling process of paddy, as shown in Figure 5a, and rice husk ash was obtained after firing at 950 °C, as seen in Figure 5b. The husk is the part of the rice in the form of a dry, scaly sheet that protects the endosperm and embryo inside. The estimated weight of husks is 20% of the gross weight of rice. The cross-section shows the husk’s main parts: lemma, palea, caryopsis, rachilla, and barren lemmas. These endosperms consist of the aleurone layer and the starchy endosperm or inner endosperm inside the caryopsis. The aleurone layer protects the embryo and the inner endosperm [65].

In general, husks have been widely used in the biomass sector. Husks are also used as an animal feed ingredient, as their protein content is low, so only a little energy is needed for the digestive process. According to Juliano [65], husks are also widely used as a fuel, fueled by uncontrolled or controlled air, distillation, pyrolysis, or a gas technique using chemicals, thermochemistry, and biochemical processes.

#### 3.1.2. Rice Straw and Rice Straw Ash

Rice straw consists of stems, leaf sheaths, leaf blades and panicles. Straw length is roughly 1–2 m, and can exceed 7 m if immersed in a deep-water reservoir [65]. Figure 6 shows the raw form of rice straw (a) and rice straw ash after firing at 950 °C (b). Straw canes are composed of an epidermal layer, an outer layer (cortex), and tissue. Parenchyma cells are parts of cells in the outer layer (outer) and are concentrated by starch granules. The moisture content in the straw after harvest is 55–70%. Protein, fat, carbohydrates, lignin, cellulose, and silica are the main components in straw [65]. Table 4 describes the characterization and structural analysis of the husk and straw.

Figure 7 shows the results of the TG curve for rice husk and rice straw. Generally, there are three significant changes in the TG curve of rice husk and rice straw—namely, in the first curve, from room temperature to 100 °C, the second curve in the temperature range between 200 and 300 °C, and the third curve in the temperature range between 300 and 550 °C; at a temperature around 550 °C upwards, the curve becomes constant. On the first curve, TG for rice husk and rice straw shows moisture release in the rice straw that occurs at temperatures up to 100 °C. The second curve, in the temperature range of 200–300 °C, shows the decomposition phase for hemicellulose, followed by cellulose. Finally, at a temperature of 300–500 °C, the third curve occurs with the decomposition of cellulose material, followed by lignin, and the curve then becomes constant. The remaining materials are silicon dioxide and oxides that do not decompose at a temperature of 1000 °C. The mass loss of rice husk and rice straw decreased most significantly in the temperature range between 200 and 600 °C, with a loss of 62% for rice husk and 67% for rice straw. Once the temperature reaches 600 °C, the mass loss for both becomes constant, as all of the material has decomposed completely. The total mass loss during the firing of rice husk and rice straw up to a temperature of 1000 °C was 75% and 80%, respectively. The material that remains after firing at a temperature of 1000 °C is called ash. The weight of the resulting ash is approximately the same as in Table 4.

According to Worasuwannarak et al. [66], Alam et al. [68], and Teh et al. [69], the mass loss of rice husk when burned is the total loss of hydrogen gas (H_2_), water (H_2_O), methyl compounds (CH_4_), carbon monoxide gas (CO), and carbon dioxide gas (CO_2_).

## 4. Materials and Methods of Wollastonite Preparation

### 4.1. Materials

Several materials were chosen as silica sources from agricultural waste—such as rice husk ash (RHA), rice straw ash (RSA), corn straw ash, wheat straw ash, and sugarcane ash—to produce wollastonite. Rice husk ash is the preferred choice as a silica source, influenced by several factors, such as its low cost and easy availability, alongside having the highest silica content (85–95 wt.%) compared to other agricultural waste products [57]. Table 5 shows the raw materials and methods used on rice husk ash (RHA) and rice straw ash (RSA) as the main sources for silica to produce the wollastonite.

### 4.2. Method

In this review, several techniques have been studied and summarized for producing wollastonite from agricultural waste, such as autoclaving, solid-state reaction, melt-quenching technique, ball milling, and sol–gel, as shown in Table 5.

#### 4.2.1. Autoclaving

An autoclave is a pressurized chamber commonly used to carry out industrial processes that require different temperatures and pressures from environmental air pressure [87]. The autoclave is used in medical applications for sterilizing purposes, and in the chemical industry to keep layers of equipment was coated against exposure to various chemicals [87].

Ismail et al. successfully prepared a β-wollastonite through an autoclaving technique utilizing rice husk ash and rice straw ash as precursors [20,21,24,71,72,79,88]. The mixing ratio of silica and calcium oxide (SiO_2_:CaO) used was 45:55, using SiO_2_ from rice husk ash or rice straw ash and CaO from local limestone. To obtain a silica and calcium oxide mixture, it was first dissolved in 100 mL of distilled water and stirred using a glass rod for 10 min before autoclaving at 135 °C for 4 and 8 h, respectively, with a pressure of 0.24 MPa. The white precipitate in water that resulted was dried in an oven at 90 °C for one day before being crushed into powder with an agate mortar. The dried powder was sintered for 1–2 h at 950 °C. XRD results exhibit the single phase of β-wollastonite after being autoclaved for 8 h and sintered for 2 h. Farah ‘Atiqah et al. [76] also used the same raw materials and method, but the sintering temperature was 1250 °C for 1 h to produce the single phase of pseudo-wollastonite.

#### 4.2.2. Solid-State Reaction

A solid-state reaction involves a substance in a powder form undergoing a long period of heat treatment and a high temperature to increase atoms’ dispersion throughout their respective solid precursors [89]. The results using this technique produce materials that are stoichiometric and are crystallized well [90]. Several previous studies also used solid-state reaction techniques to produce wollastonite using chemicals, such as the studies by Nizami and Iqbal [91], Nour et al. [92], Rashita et al. [93,94], and Shukur et al. [95].

In a study by Choudhary et al. [22], wollastonite was synthesized using a solid-state reaction in which extracted silica and eggshell powder were combined in a 1:1 stoichiometric ratio. Before thermal treatment, the powdered mixture was pelletized, and then placed in a crucible to be calcined at temperatures from 800 °C to 1300 °C for 6–9 h. After calcination at 1100 °C, single-phase wollastonite was obtained. In the lower calcination below 1000 °C, other minor phases—such as hatrurite (Ca_3_SiO_5_), larnite (Ca_2_SiO_4_), and rankinite (Ca_3_Si_2_O_7_)—were observed.

This method was also suitable to produce high-temperature wollastonite, which is also known as pseudo-wollastonite. Hossain et al. [23] synthesized pseudo-wollastonite using calcined eggshell (purity ~99% CaO) and rice husk ash extracted (purity ~99% SiO_2_) with a stoichiometric (1:1 M) ratio. Wet milling using water was carried out in a planetary ball mill for 2 h at 600 rpm to homogenize the mixture. The wet mixture was dried for 5 h at 110 °C; after that, the powder was uniaxially hydraulic pressed into discs, at a pressure of 200 MPa. The green discs were then sintered at 1200 °C for 4 h. XRD results show that the sintered powder at 1200 °C has a mixed pseudo-wollastonite phase as a major phase and parawollastonite as a minor phase.

#### 4.2.3. Melt-Quenching Technique

The melt-quenching technique is a widely used process for manufacturing bioactive glass [96,97,98,99]. Melting of oxides of calcium, silica, sodium, and phosphate starts from 1300 to 1600 °C in a platinum or alumina crucible, followed by quenching in a graphite mold to obtain rods or monoliths, or in water to obtain the frit form [96]. A ball mill is used to grind the products into powder form. The mixture is then melted in a furnace and poured into molds to create rods/cylinders or any other desired forms [96].

Hossain et al. [23] also prepared the wollastonite glass-ceramics via melt-quenching technique for comparison with the solid-state reaction method. The calcined eggshells and rice husk ash with a molar ratio of CaO:SiO_2_ to 1:1 was wet-milled for 2 h at 600 rpm using water as a medium. The wet-mixed slurry was then poured into a 30-mL platinum crucible and melted at 1400 °C for 2 h in a furnace’s air atmosphere. Next, the molten glass was then rapidly poured into distilled water, yielding a transparent glass. The wollastonite glass was then crushed and sieved through a 60-μm sieve to obtain a homogeneous glass powder. The glass powder was mixed with 3 wt.% of PVA as a binder and pressed at a pressure of 200 MPa. The pressed pellets were then sintered at 1200 °C for 1 h. XRD analysis shows a purely amorphous phase for 1400 °C, and after heat treatment at 1200 °C, the amorphous phase turns to a crystalline peak for pseudo-wollastonite, with JCPDS no. 74-0784.

Shivani and Kulvir (2020) [25] produced bioactive calcium silicate glass using a melt-quenching technique. The precursors were from sugar bagasse and corn for the silica source and eggshells for the CaO source. The sugar bagasse and corn were fired at 500 °C for 5 h, while the eggshells were calcined at 1000 °C for 2 h. The calcined powder was mixed using an agate mortar pestle for 2 h to homogenize the mixture. Next, the mixture was melted at 1550 °C with a soaking time of 30 min in a muffle furnace. The molten glass was then quenched in the air onto copper plates. The frit samples were crushed into powder form for characterization. The XRD results show that the peak is in the amorphous phase for calcium silicate glass.

#### 4.2.4. Sol–Gel

In the early 1980s, sol–gel techniques were introduced for Hayashi and Saito’s glass synthesis [100]. This method is based on the polymerization reaction of hydrolysis and repeated condensation between silicon alcoholates and metal alcoholates or metal salts suitable for producing inorganic gels with the following networks:
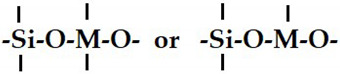

where M is a metal atom, such as titanium, zirconia, aluminum, boron, etc.

Hayashi and Saito [100] pioneered the production of calcium silicate through sol–gel techniques using calcium ethylate (Ca(Oet)_4_)- and ethylic silicate (Si(Oet)_4_)-based materials, referring to the phase diagram of the CaO–SiO_2_ binary system to produce the calcium silicate phase. The starting material solution was stirred and mixed with ethanol and argon gas for 4 h. The color of the solution changed from yellow to brown, and the solution was left for the gelatinization process to take place. Transparent glass with the composition of CaO·9SiO_2_ and CaO·4SiO_2_ was successfully produced after hydrolysis at ambient humidity, and slowly burned at a temperature of 800 °C.

Calcium nitrate (Ca(NO_3_)_2_) and TEOS are commonly used as raw materials in the production of calcium silicate by sol–gel techniques [101,102,103]. The sol–gel process yields higher purity, better surface area, and greater homogeneity than the fusion method [100,101]. One benefit of this process is that glass can be prepared at lower temperatures without reducing the Na_2_O component, which aims to reduce the melting temperature without changing the glass composition [101].

Palakurthy et al. [70,73] used a modern sol–gel technique to synthesize calcium silicate; the raw materials for the synthesis of the calcium silicate were derived from rice husk ash and eggshell. Firstly, rice husk was washed and fired at 600 °C for 4 h to obtain rice husk ash for the silica source. Subsequently, rice husk ash in 2 mol/L of NaOH solution was heated to obtain a sodium silicate solution. For the calcium oxide source, the eggshells were collected, washed, and then dried. Next, the dried eggshells were then crushed into a fine powder before being calcined at 900 °C for 2 h. CaO obtained from the eggshell was dissolved in 2 mol/L hydrochloric acids to produce a calcium chloride solution. The sodium silicate solution was then continuously combined with the calcium chloride solution. It was possible to obtain a white, jelly-like suspension. In a sealed bottle, the gel was allowed to age for three days at room temperature. The gel was then washed three times with deionized water before drying at 70 °C for 24 h, followed by a further 6 h at 120 °C. Sintering at 800 and 900 °C yielded calcium silicate powders. The XRD peak pattern exhibited an identical trend to that of the wollastonite and was well balanced with JCPDS no. 840655.

Saravanan et al. [78] and Azeena et al. [83] also synthesized calcium silicate or wollastonite via the sol–gel method, utilizing rice straw ash and calcium nitrate as raw materials. Rice straw was washed to eliminate any impurities such as clay and dried at 60 °C. The dried straw was finely chopped, treated with 1 N HCl under boiling conditions for 1 h, and washed to neutral pH. Finally, the acid-treated straw was calcined at 550 °C for 5 h to yield white rice straw ash containing silica. White rice straw ashes were dissolved in 0.8 M NaOH for 1 h under boiling conditions to yield sodium silicate solutions. The solution was filtered to obtain a clear filtrate. Next, an equmolar solution of calcium nitrate in distilled water was prepared. Sodium silicate solution was slowly and continuously added dropwise into the calcium nitrate solution. White precipitates of the calcium silicate were obtained via co-precipitation. The particles were separated by centrifugation, washed several times with ethanol and ultra-pure water, and then dried at 60 °C to yield wollastonite particles. The peak of these wollastonite particles matches with JCPDS no. 043-1460, corresponding to the wollastonite phase [78]. Meanwhile, Azeena et al. [83] obtained the crystallinity and amorphous nature of wollastonite with the 2θ values exhibited at 22.36°.

#### 4.2.5. Milling

The milling process is a mechanical crushing process. Ball milling is the most used technique in laboratory-scale studies. The milling balls are small balls that move freely in the milling media and produce forces due to collisions between the balls and the milling media, causing changes to the milling sample. It has been found that some changes occur, such as particle breakdown, flattening, and merging [104]. There are two types of milling conditions carried out—namely, dry and wet. Wet milling involves milling in aqueous media, such as water and ethanol. The final size of the product depends on the techniques used, media characteristics (e.g., material, shape, and size), environment (e.g., dry, in aqueous or non-aqueous medium, with or without dispersing agent) and load ratio (media weight to powder weight).

Phuttawong et al. [81] produced wollastonite from snail (*Pomacea canaliculata*) shell and rice husk ash. The snail shell was washed with water and dried in an oven at 100 °C for 24 h, while the rice husk ash was also dried in an oven at 100 °C for 24 h. Both precursors were manually crushed and sieved through a 60-mesh sieve. The shell powder was calcined at 800 °C for 2 h in a furnace, with a mixture of shell powder and rice husk ash in a molar ratio of 1:1. Alumina balls were used to dry mill the mixed solids for 5, 6, 7, and 8 h. The milled powders were then calcined at 800, 900, and 1000 °C for 2 h in a furnace. The XRD results concluded that the calcium silicate pattern structure corresponded to JCPDS no. 27-0088, which was not affected by increasing the milling time. The intensity of the XRD peaks also increased and did not change the phase transformation of calcium silicate, when the milling time was prolonged.

### 4.3. Advantages and Limitations of the Processing Methods

Each method of synthesis of bioceramic materials has advantages and limitations. Table 6 shows the brief virtues and drawbacks of the methods used in the production of bioceramic materials.

## 5. Bioactivity and Biocompatibility Properties

In the field of biomaterials, several significant tests and analyses need to be verified in order to determine whether or not the material is suitable as a biomaterial. The initial tests that need to be done include bioactivity and biocompatibility tests, to determine whether the material can form an apatite layer with living tissue, for example. Meanwhile, biocompatibility tests are performed to determine whether the biomaterials are nontoxic and harmful to living tissue. Table 7 shows other researchers’ results on the bioactivity and biocompatibility testing for wollastonite using rice husk ash and rice straw ash as raw materials.

### 5.1. Bioactivity Properties

Bioactivity can create a chemical bond with living tissues, recognized by the capacity to stimulate apatite formation on the scaffold surface upon immersion in a simulated body fluid (SBF) in vivo, in vitro, or ex vivo [110]. In vivo evaluation is performed by inserting the sample into the animal. In contrast, in vitro evaluation is carried out in a laboratory environment, by soaking the sample in the SBF solution. The pH and ions in the solution are equal to the solution in the human body. The ex vivo technique, on the other hand, uses a sample of cells removed from animals or humans and tested in a laboratory environment (in vitro) [89]; this technique is also known as an in situ technique. Usually, a substance’s bioactive properties are determined using in vitro techniques, before being introduced into animals through in vivo techniques.

#### 5.1.1. The Apatite Formation Mechanism for the CaO–SiO_2_ System In Vitro and In Vivo

Apatite is a phosphate mineral group with the chemical formula Ca_5_(PO_4_)_3_(F, Cl, OH) [111] Apatite usually refers to hydroxyapatite (Ca(PO_4_)_3_OH), fluorapatite (Ca_5_(PO_4_)_3_F), or chlorapatite (Ca(PO_4_)_3_Cl), and is widely found in igneous, metamorphic, and sedimentary rocks [112]. Hydroxyapatite is a major component of tooth enamel, and is widely used for bone applications [113].

In CaO–SiO_2_ silicate systems for ceramics, glass, and glass-ceramics, apatite formation occurs when this material is immersed in SBF solution. This apatite-formation mechanism involves ions calcium exchanging from glass or glass-ceramics with phosphate ions in the SBF solution. There are four steps in the mechanism of apatite layer formation for the CaO–SiO_2_ system, as shown in Figure 8 and described as follows:First step:

The first effect of immersing silicate glass samples in SBF solution is the release of ions from the silicate glass samples. Due to the variable solubility of Ca^2+^ ions and SiO^4+^ ions from silicate glass, Ca^2+^ ions are favored for release into SBF solution over SiO^4+^ ions. The release of Ca^2+^ ions allows the hydrolysis of silica groups in silicate glass samples, resulting in the formation of a silicon-rich layer (Si–OH) on the silicate glass samples’ surface [94], as in Equation (1):Si–O^−^ + H^+^ + OH^−^→Si–OH^+^ + OH^−^(1)

Second step:

At this point, the SBF solution’s pH rises as H^+^ ions in the solution are replaced by Ca^2+^ ions from silicate glass samples. Ca^2+^ ions in the SBF solution are electrostatic, which means that they are quickly attracted to the newly formed, negatively charged, silanol-rich layer. Ca^2+^ ions dissolve in and exchange with H^+^ ions in the SBF solution, interfering with the CaO–SiO_2_ ceramic network, as shown in Equation (2). At this stage, the SBF solution’s supersaturation concentration rises.
Si–O–Si + H_2_O→SiOH + OH–Si(2)

Third step:

As a result of the electrostatic attraction to the negatively charged surface of the silicate glass sample by Si–O–Si, Ca^2+^ ions begin to bind to the surface of the silicate glass sample, accompanied by PO_4_^3−^/HPO_4_^2−^ adsorption on the surface of the Ca^2+^ ions. The transfer between Ca^2+^ and PO_4_^3−^/HPO_4_^2−^ ions on the surface of the SiO_2_-rich layer forms the Ca–P_2_O_5_ layer, followed by the amorphous growth of CaO–P_2_O_5_, which is a combination of soluble ion calcium and phosphate substances derived from SBF solution.

Fourth step:

This last step is the formation of a Ca–SiO_2_-based apatite layer on the surface of the ceramic sample. During the process of apatite formation, there is a change in the ceramic sample’s weight, the pH value, and the SBF solution’s ion concentration. These factors influence one another in the reaction of the apatite formation mechanism.

Figure 9 briefly shows the bioavailability mechanisms for silicate glass ceramics using immersion techniques in SBF (in vitro) solution or incorporated into animal bones (in vivo). The first five steps are the same through different medium uses; in the in vitro technique, the sample is immersed in SBF solution, while in the in vivo technique, the sample is inserted into the animal.

#### 5.1.2. SBF

The attempts to define a relationship between in vitro and in vivo studies have created research solutions that mimic human cellular fluid. Some researchers have used TRIS buffer alone to test ceramics, glass, and glass-ceramic surfaces to form apatite [97,114,115], as is acknowledged by biological laboratories. The solutions as prepared by Kokubo and Takadama [116] constituted a critical point in the preparation of solutions simulating in vivo environments, with an ion concentration close to that of human blood plasma, as shown in Table 8.

Since SBF is supersaturated in apatite, improper preparation can cause apatite precipitation in the solution. Each time the SBF is prepared, maintain the prepared solution colorless and transparent, and ensure no deposits on the container’s surface. If precipitation occurs, discontinue SBF preparation, discard the solution, resume washing the apparatus, and re-prepare the SBF.

First, to make 1 L of SBF, pour 700 mL of deionized water into a 1000-mL plastic beaker and stir with a magnetic stirring bar. Cover the beaker with a watch glass or plastic wrap and place it in the water bath. Under stirring, heat the water in the beaker to 36.5 ± 1.5 °C. Only the 1st–8th order reagents should be dissolved—one by one, in order—into the solution at 36.5 ± 1.5 °C, as shown in Table 9. The 9th (Tris) and 10th (small amount of HCl) order reagents are dissolved in the pH modification process, as follows:

Following that, each sample is soaked in the SBF on a predetermined day at a pH of 7.4 and a temperature of 36.5 °C. Using the following formula, the SBF volume needed to soak the cylindrical or pellet sample can be calculated: V_s_ = S_a_/10, where V_s_ is the volume of SBF (mL), and S_a_ is the specimen’s apparent surface area (mm^2^). Every three days, the SBF should be refreshed. Following the soaking duration, rinse the samples in acetone for 2 h, rinse three times with deionized water to eliminate buffer salts, dry in an incubator for 24 h, and characterize the samples.

#### 5.1.3. Bioactivity Studies

The analysis of apatite formation on a material in simulated body fluid (SBF) is a widely accepted method for predicting a material’s in vitro bone bioactivity. Table 10 briefly described the summary of in vitro studies on the bioactivity of wollastonite derived from RHA and RSA from other researchers.

Table 10 summarizes how all wollastonite samples from agricultural waste—especially RHA and RSA—have excellent bioactivity properties. Apatite was discovered after just 3 days of immersion in the SBF solution. This demonstrates that agricultural waste silica has equivalent or superior properties to commercial silica material. As a result, we could conduct a deeper study on the medical applications of wollastonite from agricultural waste, such as its biocompatibility properties.

### 5.2. Biocompatibility Properties

A simple description of the biocompatibility of materials is the ability to avoid opposing tissue reactions. Another important biocompatibility concept is a material’s capacity to achieve a suitable host response in detailed applications [117,118]. This interpretation helps connect material behavior or characteristics with results, i.e., biological necessities, a particular request, a detailed medical device, or a biomaterial used as a biomedical implant [119].

#### 5.2.1. In Vitro Biocompatibility

For a broad range of biomaterials used in medical instruments and prostheses, in vitro cytotoxicity assays are the first biocompatibility screening studies. Further application-specific tests are conducted to determine biomaterials’ biocompatibility under end-use conditions, regarding a candidate biomaterial’s cytotoxicity profile. Mainly, biomaterials found by in vitro studies to be nontoxic would not be toxic in in vivo tests. More research must be conducted on biomaterials found to be toxic or harmful by in vitro assays in order for them to be clinically acceptable. There are examples of toxicity with low biomaterial levels, but this does not restrict or prevent their use in biomedical devices. For example, glutaraldehyde-fixed porcine valves induce adverse effects in vitro due to the slight remnants of glutaraldehyde. Despite being harmful in in vitro results, this material is still used in prosthetic heart valves’ clinical use. This is an example of where the therapeutic benefit of using this material compensates for the chance of minimal levels of toxicity.

The most accepted form of biocompatibility test is the use of cell culture systems to detect cytotoxicity, cell activation, cell adhesion, or cell death. The standard assays used in biocompatibility are cell culture assays to study new biomaterials, and to perform biocompatibility evaluations for requirements such as biomaterials in general, medical devices, and prostheses [120]. Three kinds of cell culture experiments are widely used to determine biocompatibility: dilution extract, diffusion of agar, and direct contact. Direct-contact cell culture is the approach most frequently used by researchers to determine the biocompatibility of new biomedical devices or biomaterials. In this study, the researcher may use the type of cell for which the biomaterial under evaluation is targeted for clinical use.

#### 5.2.2. MTT Assay

Tim Mosmann introduced the well-documented MTT assay in 1983 [121]. The proliferation, viability, and cytotoxicity of cells can be determined using this colorimetric assay. This assay is beneficial because it can be performed quickly on a microtiter plate assay and read on an ELISA plate reader on a spectrophotometer at 570 nm [121]. Altmann 1977 used light and electron microscopy to demonstrate oxidative and non-oxidative enzymes’ activities in mitochondria, as stated by Inayat et al. [119]. The reduction of tetrazolium salts to their corresponding formazan precipitates was used for the histochemical demonstration of oxidative and non-oxidative enzymes’ activities in mitochondria. Newly, tetrazolium MTT is decreased by the tetrazolium ring’s cleavage by succinate dehydrogenase in the mitochondria, in an insoluble purple formazan reaction (Figure 10). The formazan gathers within the cell, since it cannot move over the cell membrane.

Furthermore, to the spectrophotometric-grade dimethyl sulfoxide (DMSO), isopropanol, or another appropriate solvent, the formazan is solubilized and released. Therefore, it becomes easily measured colorimetrically. Cytotoxic concentration is usually measured by the concentration that kills 50% of the cells, usually recognized as LC_50_.

#### 5.2.3. In Vivo Biocompatibility

An in vivo test of the biocompatibility of medical and biomaterials devices is performed in order to ensure that the application works as proposed and does not cause any harm to the user or patient. In vivo biocompatibility testing aims to identify whether a medical device presents a risk of harm or trauma to the patient, by testing under clinically relevant conditions. Government agencies—that is, the FDA; and the regulatory bodies, ASTM and ISO—have recently made comprehensive efforts to include guidelines, standards, protocols, and procedures. This can be applied in the in vivo evaluation of the compatibility of tissues with medical devices. This formulates an interesting dealings with the ISO 10993 standard—Biological Evaluation of Medical Devices—by providing a systematic framework for the in vivo tissue compatibility assessment of medical devices. To simulate end-use applications, in vivo biocompatibility analyses are chosen. Medical devices and their constituent biomaterials may be categorized depending on the character and duration of the medical device’s body contact to help select appropriate tests. The tissue contact groups and subgroups, as well as the contact length groups, were derived from previous requirements, procedures, and medical device safety evaluation regulations. Some devices can fit into more than one classification, in which case testing relevant to each classification should be considered.

#### 5.2.4. Biocompatibility Studies

According to Table 11, all wollastonite samples from agricultural waste—especially RHA and RSA—have excellent biocompatibility properties. Apatite was discovered after just 3 days of immersion in the SBF solution. This demonstrates that agricultural waste silica has equivalent or superior properties to commercial silica material. Thus, a deeper study on the biomedical and biomaterials application of wollastonite from agricultural waste, must go for some characterization such as an antibacterial property and some clinical testing on animals

## 6. General Overview and Future Perspectives

Current studies regarding wollastonite or calcium silicate from agricultural waste products (RHA and RSA) as potential dental and bone substitutes have been successfully conveyed. Wollastonites from RHA and RSA have been increasingly studied for their fundamental properties—such as bioactivity and biocompatibility—because of these materials’ combinatory properties for the compulsory requirements of biomedical implants. The formerly cited summary of studies on the bioactivity and biocompatibility of wollastonite from RHA/RSA is summarized in Table 10; Table 11, respectively. Wollastonite is one of the most researched bioceramics for synthetic bone replacements in implants and dentistry, owing to its high bioactivity and biocompatibility.

Hopefully, the commercialization of wollastonite products based on RHA/RSA will occur. It should also be noted that most of the attempts at wollastonite production at a lab scale have been successful, and the following studies should be performed at large scales. More investigation should be investigated in order to secure sustainable wollastonite production from silica-rich agricultural residues. The number of in vivo bioactivity and biocompatibility studies involving wollastonite RHA/RSA is increasing; however, most of these are conducted in vitro on cell cultures from embryonic or lower-order lifeforms.

A critical target for future wollastonite bioceramics research using RHA/RSA is to strengthen mechanical properties in order to determine the long-term effects of wollastonite derived from RHA/RSA in humans, allowing for the design of strength and in vivo studies with definable endpoints. The recently developed wollastonite mentioned above is well suited for clinical research and may lead to new insights into novel biomedical implants made from agricultural waste.

Finally, we hope that future concerns about the health risks and benefits of RHA/RSA in biomedical applications will be guided by strong scientific evidence generated from carefully designed studies using appropriate methods. Such research is crucial to ensuring that agricultural-waste-enhanced goods’ ultimate regulation is focused on reality, rather than fear or prejudice.

## 7. Conclusions

Agricultural waste—mainly rice husk ash (RHA) and rice straw ash (RSA)—has been used as feedstock in the manufacture of bio-based goods and, eventually, become a new source of renewable energy in many areas of life. Its use could minimize pollution, though still having an environmental effect due to its dependence on chemical products. It could be used as a new raw material in biomedical applications, further enhancing its importance. In biomedical applications, rice husk ash produced by firing rice husk is primarily useless and of no value. Rice husk ash contains a high yield of silica, making it an excellent source for wollastonite. Though rice husk ash is not the first sustainable source of wollastonite synthesis, it does provide a low-cost, renewable, and affordable source for apatite formation. Chemical analysis shows that these materials function similarly to wollastonite obtained via other methods; they have also shown excellent properties in bioactivity and biocompatibility; and the ability to perform in the desired manner.

## Figures and Tables

**Figure 1 materials-14-05193-f001:**
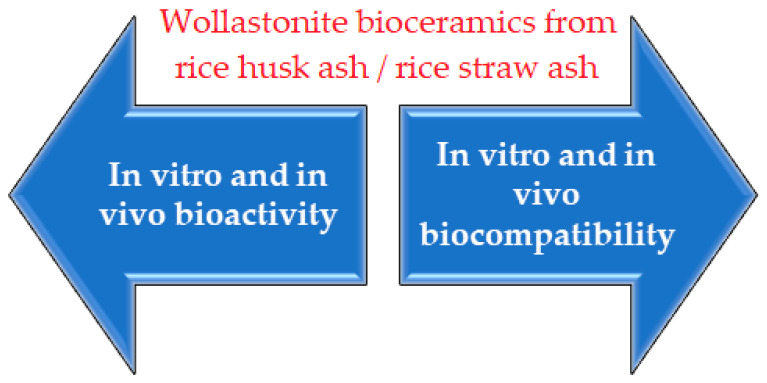
The core outline of this review.

**Figure 2 materials-14-05193-f002:**
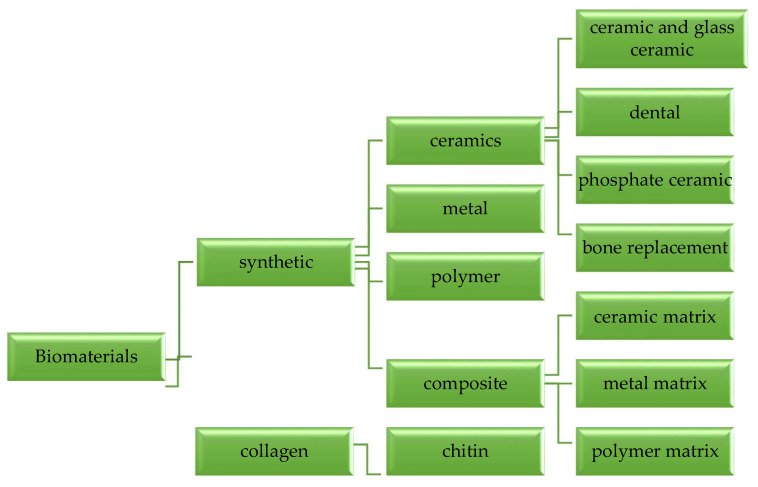
Classification of natural and synthetic biomaterials.

**Figure 3 materials-14-05193-f003:**
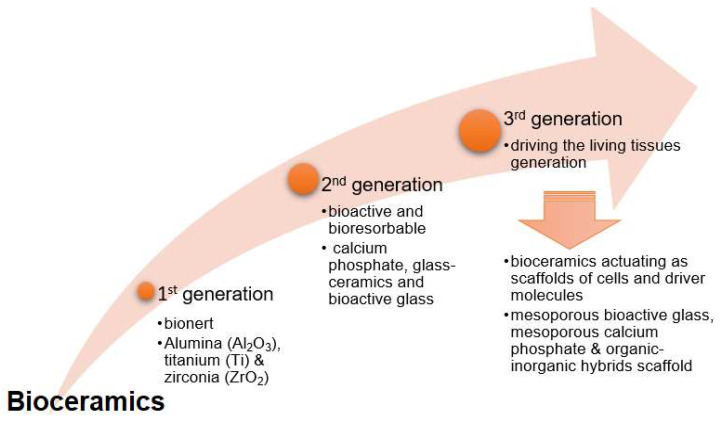
A schematic layout of the three generations of bioceramics [34].

**Figure 4 materials-14-05193-f004:**
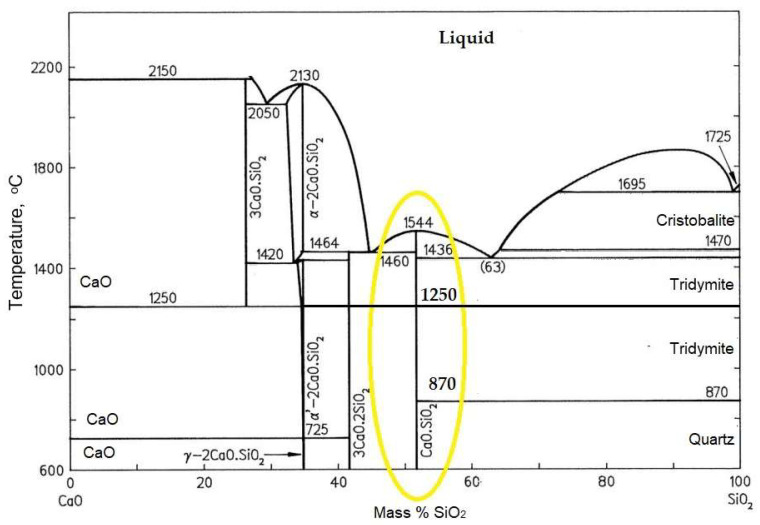
Phase diagram of the CaO–SiO_2_ system.

**Figure 5 materials-14-05193-f005:**
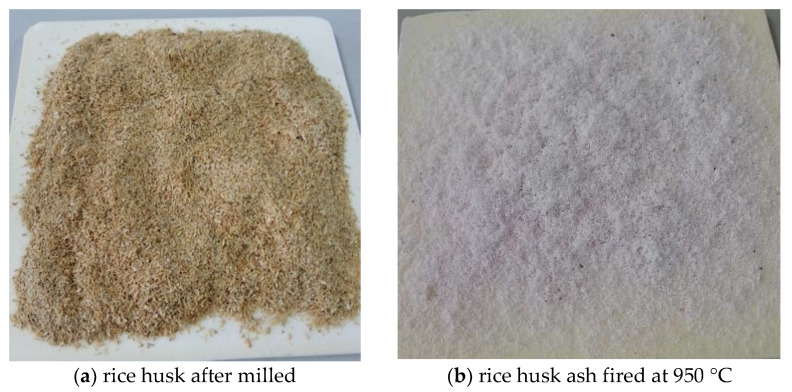
(**a**) Rice husk after milled and (**b**) rice husk ash after firing at 950 °C.

**Figure 6 materials-14-05193-f006:**
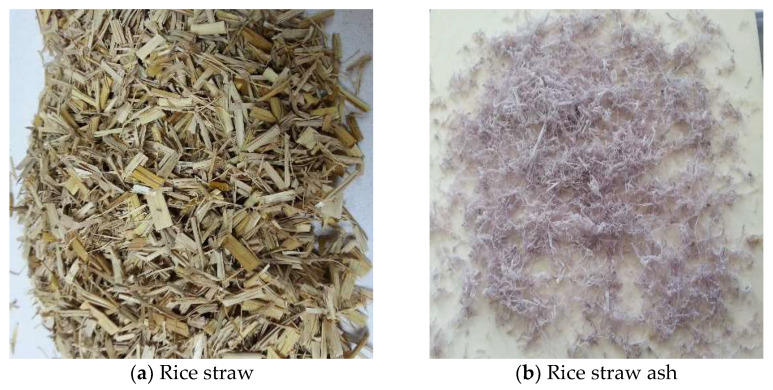
(**a**) Rice straw and (**b**) rice straw ash after firing at 950 °C.

**Figure 7 materials-14-05193-f007:**
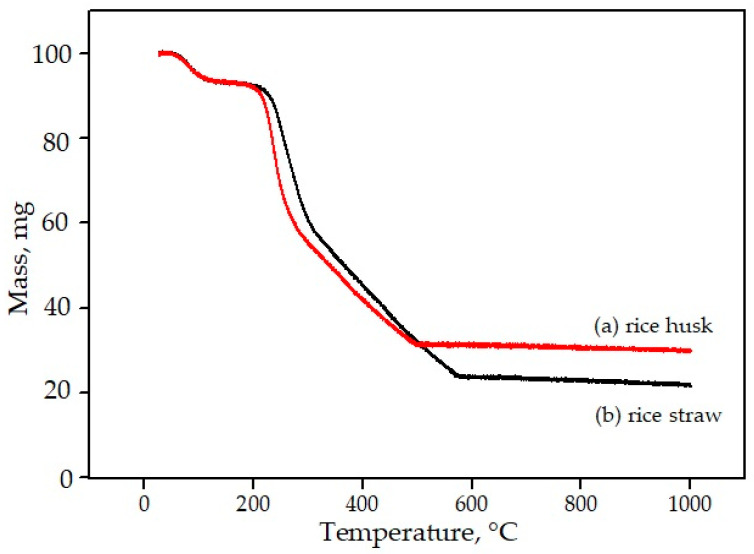
TG curve for (**a**) rice husk and (**b**) rice straw after firing at 1000 °C.

**Figure 8 materials-14-05193-f008:**
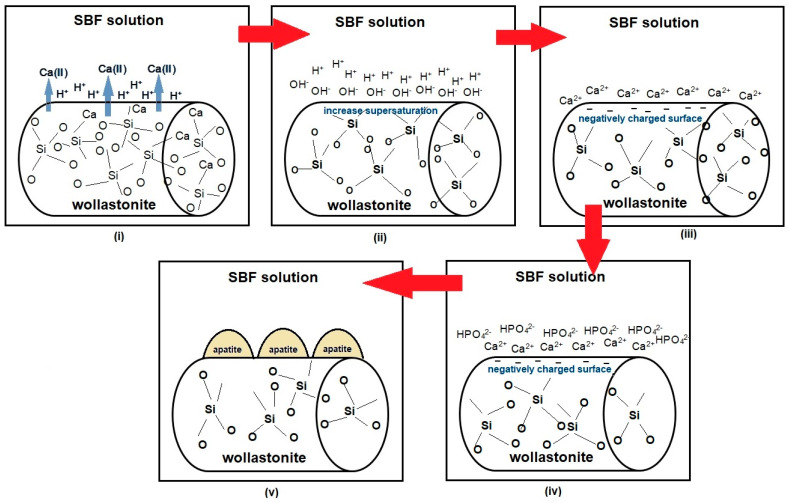
The processes of the apatite formation mechanism on the CaO–SiO_2_ ceramics in SBF.

**Figure 9 materials-14-05193-f009:**
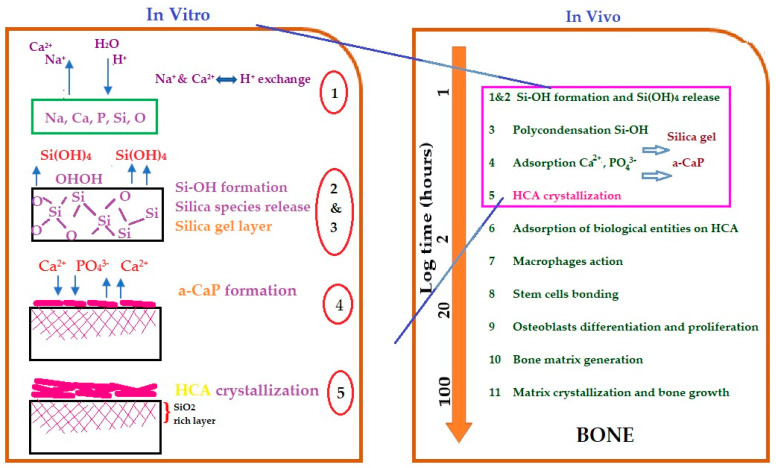
Bioactive mechanisms of silica glass ceramic samples using in vitro and in vivo techniques. (HCA = apatite hydroxycarbonate). Modified from Vallet-Regi [89].

**Figure 10 materials-14-05193-f010:**
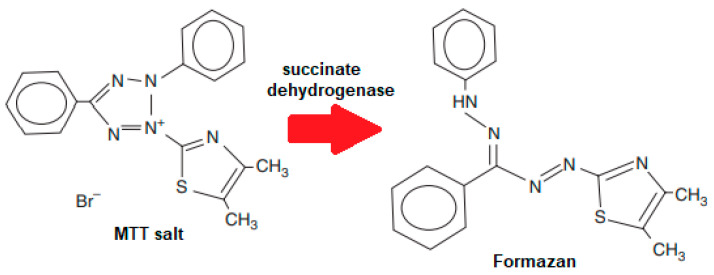
Reduction of MTT salt to purple formazan.

**Table 1 materials-14-05193-t001:** Comparison of the structure and structural parameters of adult human calcified tissues’ inorganic phases [31,32].

Composition, wt.%	Bone	Hydroxyapatite	Enamel	Dentin
Calcium, Ca	34.8	39.6	36.5	35.1
Phosphorus, P	15.2	18.5	17.7	16.9
Ca/P molar ratio	1.71	1.67	1.63	1.61
Sodium, Na	0.9	-	0.5	0.6
Magnesium, Mg	0.72	-	0.44	1.23
Potassium, K	0.03	-	0.08	0.05
Carbonate, CO_3_^2−^	7.4	-	3.5	5.6
Fluoride, F	0.03	-	0.01	0.06
Chloride, Cl	0.13	-	0.30	0.10
Pyrophosphate, P_2_O_7_^4−^	0.07		0.022	0.10
Total inorganic	65	100	97	70
Total organic	25	-	1.5	20
Water	10	-	1.5	10
Ignition products (800 °C)	HA + CaO	HA	Β-TCP + HA	Β-TCP + HA
Elastic modulus (GPa)	0.34–13.8	10	80	15
Tensile strength (MPa)	150	100	10	100

**Table 2 materials-14-05193-t002:** The physical and chemical properties of wollastonite.

Description	Value
Color	White
Luster	Vitreous, Pearly
Molecular weight, gmol^−1^	116
Specific gravity, gcm^−3^	2.86–3.09
Refractive index	1.63
pH (10% slurry)	9.9
Solubility in water (g/100 cc)	0.0095
Density (g/cm^3^)	2.70–3.00
Hardness (Mohs)	4.5–5
Melting point (°C)	1540

Source: www.mindat.org/min-4323.html (accessed on 6 April 2021).

**Table 3 materials-14-05193-t003:** Composition of agricultural waste.

Agricultural Waste	Chemical Composition (% *w*/*w*)			Ash (%)	References
	Cellulose	Hemicellulose	Lignin		
Rice husk	32.7	31.7	18.8	16.3	[14,57]
Rice straw	41.9	25.6	0.8	16.5	[58,59]
Palm oil trunk	39.9	21.2	22.6	1.9	[60]
Palm oil frond	31.5	19.2	14.0	12.3	[61]
Sugarcane bagasse	30.2	56.7	13.4	1.9	[62]
Corn stalks	42.7	23.3	17.5	9.8	[59]
Wheat straw	32.8	38.0	8.9	1.4	[63]
Soy stalks	34.5	24.8	19.8	10.4	[62]

**Table 4 materials-14-05193-t004:** Characterization and structural analysis of the husk and straw [66,67].

Parameters	Husk	Straw
Origin	Rice milling	After harvesting
Quantity	20–22% from rice weight	2–8 ton/ha
Moisture content, (%)	10	60 (wet weight) 10–12 (dry condition)
Density, (kgm^−3^)	100–150 200–250 (on land)	75 (loose straw) 100–180 (compact form)
Carbohydrate components, (%) (average)	Cellulose: 28–36 Hemicellulose: 12 Lignin 9–20 Gross protein: 1.9–3.0	Cellulose: 24–34 Hemicellulose: 19–29 Lignin: 5–11 Gross protein:2.8–4.4
Caloric content, (MJkg^−1^)	14–16 (10% moisture content)	14–16 (14% moisture content)
Ash, (%)	13.2–21.0	10.4–21.8
Silica, (mgg^−1^)	18.8–22.3	11–15
Calcium, (mgg^−1^)	0.6–1.3	0.9–5.0
Phosphorus, (mgg^−1^)	0.3–0.7	0.61–0.65

**Table 5 materials-14-05193-t005:** Summary of rice husk ash (RHA) and rice straw ash as raw materials, and the methods used to fabricate wollastonite.

Raw Material	Method	References
Rice husk ash and limestone	Autoclaving	Ridzwan et al. [24]
Eggshell and rice husk ash	Solid-state reaction	Choudhary et al. [22]
Rice husk ash and eggshells	Solid-state reaction	Hossain et al. [23]
corn, sugarcane and eggshells	melt quench technique	Shivani and Kunjir [25]
Rice husk ash and eggshell	Sol–gel	Palakurthy et al. [70]
Rice husk ash and eggshell	Autoclaving	Ismail et al. [71]
Rice straw ash and limestone	Autoclaving	Ismail et al. [72]
Rice husk ash and eggshell	Sol–gel	Palakurty et al. [73]
Rice husk ash and eggshell	Sol–gel	Palakurthy et al. [74]
Rice husk ash and eggshells	Solid state	Sultana et al. [75]
Rice husk ash and limestone	Autoclaving	Farah ‘Atiqah et al. [76]
Rice husk ash and limestone	Autoclaving	Roslinda et al. [77]
Rice straw ash	Sol–gel	Saravanan et al. [78]
Rice husk ash and limestone	Autoclaving	Ismail et al. [79]
Rice husk	Autoclaving	Alshatwi et al. [80]
Rice husk ash and shell of a snail	Milling	Phuttawong et al. [81]
Rice husk	Autoclaving	Athinarayanan et al. [82]
Rice straw ash	Sol–gel	Azeena et al. [83]
Rice husk ash/PCL	Sol–gel	Naghizadeh et al. [84]
Rice husk ash	Sol–gel	Nayak et al. [85]
Rice husk ash	Sol–gel	Nayak et al. [86]

**Table 6 materials-14-05193-t006:** Advantages and limitations of the methods in producing bioceramics and bioglasses.

Methods	Advantages	Limitations	Ref.
Autoclaving	Very tough, able to bear high pressures and temperatures for long-term processing.	-	[105]
Solid-state	Easy control of operating conditions	The formation of toxic waste products; not suitable for mass production	[106]
Melt-quenching	- Simple and low-cost - High-temperature processing	- Reagents with high purity required - Low reproducibility of doped ions within the glass structure - Low homogeneity	[107]
Sol–gel	- High homogeneity and purity - Low-temperature processing - Ability to produce a wide range of nano-/microstructures	- Expensive raw materials - Length and complexity of the process	[90,108,109]
Milling	Non-mixed nanomaterial systems can be produced.	Contamination from milling jar	[90,104]

**Table 7 materials-14-05193-t007:** The bioactivity and biocompatibility testing for wollastonite using rice husk ash and rice straw ash as raw materials.

Ref	Bioactivity	Biocompatibility
Ridzwan et al. [24]	+	−
Choudhary et al. [22]	+	+
Hossain et al. [23]	+	+
Shivani and Kunjir [25]	+	−
Palakurthy et al. [70]	+	−
Ismail et al. [71]	+	−
Ismail et al. [72]	+	−
Palakurty et al. [73]	+	+
Palakurthy et al. [74]	+	+
Sultana et al. [75]	+	−
Farah ‘Atiqah et al. [76]	+	+
Roslinda et al. [77]	+	+
Saravanan et al. [78]	−	+
Ismail et al. [79]	+	−
Alshatwi et al. [80]	−	+
Phuttawong et al. [81]	−	−
Athinarayanan et al. [82]	−	+
Azeena et al. [83]	−	+
Naghizadeh et al. [84]	+	−
Nayak et al. [85]	+	−
Nayak et al. [86]	+	−

**Table 8 materials-14-05193-t008:** Ion concentrations of SBF and human blood plasma. Data from Vallet-Regi and Kokubo and Takadama [89,116].

Ion Concentration	SBF (mM)	Human Blood Plasma (mM)
Na^+^	142.0	142.0
K^+^	5.0	5.0
Mg^2+^	1.5	1.5
Ca^2+^	2.6	2.5
Cl^−^	148.8	103.0
HPO_4_^2−^	1.0	1.0
SO_4_^2−^	0.0	0.5
pH	7.4	7.2–7.4

**Table 9 materials-14-05193-t009:** Order, amount, and purity of reagents for preparing 1 L of SBF.

Order	Reagent	Amount (g)	Purity (%) ± 2.0
1	NaCl	8.035	99.0
2	NaHCO_3_	0.355	99.0
3	KCl	0.225	99.0
4	K_2_HPO_4_·3H_2_O	0.231	99.0
5	MgCl_2_·6H_2_O	0.311	98.0
6	1.0 M HCl	35.0	-
7	CaCl_2_	0.292	95.0
8	Na_2_SO_4_	0.072	99.0
9	Tris	6.118	99.0
10	1.0 M HCl	±10.0 mL	-

**Table 10 materials-14-05193-t010:** Summary of in vitro studies on the bioactivity of wollastonite derived from RHA and RSA.

Ref.	Bioactivity Study
Choudhary et al. [22]	The wollastonite scaffold’s XRD patterns showed that the HA process (JCPDS no. 09-0432) precipitated after 3 days of soaking. After 10 days, the apatite had fully covered the surface, indicating that the SBF was responsive. Increasing the soaking period caused apatite to appear as the main phase, with high peaks, and decreased the wollastonite peak. As a result, increasing the soaking time increased apatite deposition.
Hossain et al. [23]	After soaking in SBF, the peak of the α-wollastonite phases decreased with soaking time for both samples (α-wollastonite ceramic (WC) and α-wollastonite glass-ceramic (WGC)). This is due to the deposition of the hydroxyapatite (HA) process on the surface of WC and WGC. The characterization peak of HA at around 2θ = 32° confirms the formation of the HA process. As a result, the number of HA peaks increases as the soaking time increases from 7 to 28 days. By comparing the number of HA peaks between WGC and WC in the SBF solution, it was determined that WGC is more degradable than WC. As a result, the findings show that the WGC is more capable of forming HA than WC.
Ridzwan et al. [24]	After 7 days of soaking in SBF, hydroxyapatite (HA) was formed on the β-WI and β-WFD samples’ surfaces. At the end of the soaking period, amorphous calcium phosphate (ACP) and calcium-deficient hydroxyapatite (CDHA) were deposited on both samples’ surfaces. Because of single-phase HA formation after 21 days of soaking, β-WI was more bioactive than β-WFD.
Shivani and Kunjir [25]	The hydroxyapatite (HA) with ICDD no. 09-432 is observed when soaking in SBF for glasses derived from sugarcane leaf ash, corn husk ash, or eggshell powder. These peaks may be linked to amorphous HA. These peaks vanished later in the soaking phase, suggesting the formation of metastable HA in these glasses.
Palakurthy et al. [70]	The hydroxyapatite (HA) phase (JCPDS no: 090432) was detected in the XRD analysis, and an increasing amount of HA phase was observed with increasing soaking time. After 14 days of soaking, the wollastonite phase’s diffraction intensity almost completely disappeared, and was replaced by HA as the main phase of wollastonite derived from rice husk ash. These findings show that wollastonite ceramics made from RHA and eggshell have a faster rate of HA growth on their surface, related to their surface microstructure.
Ismail et al. [71]	All samples showed good bioactivity properties, with a thin layer of glass of amorphous calcium phosphate (ACP) on the sample surface.
Ismail et al. [72]	β-wollastonite samples dried in the incubator at body temperature were more bioactive than freeze-dried samples. Both sets of samples produced the same types of the calcium phosphate group on the surface—namely, amorphous calcium phosphate (ACP), and calcium-deficient hydroxyapatite (CDHA).
Palakurthy et al. [73]	The hydroxyapatite (HA) showed peaks characteristic of JCPDS no. 09-0432, suggesting that HA layer growth began immediately on the sample surface after soaking it in SBF solution. As the soaking time increased, the actual wollastonite phase’s diffraction intensity decreased dramatically, and was replaced by the HA phase. Because of the non-homogeneous distribution of HA on the sample’s surface, the wollastonite phase was still detectable after 21 days of soaking.
Palakurthy et al. [74]	The XRD pattern indicates the representative diffraction peaks of HA in all samples, consistent with JCPDS no. 09-0432. The diffraction strength of the initial wollastonite process was significantly reduced and replaced with HA. Furthermore, the peak intensities of HA increased with increasing exposure time in SBF solution from 7 to 21 days. It can also be shown that the intensity of these diffraction peaks increased from pure wollastonite (W) to silver-doped wollastonite (WAg), suggesting that WAg has more HA mineralization on its surface than W.
Sultana et al. [75]	As a result of immersion in SBF, the surface of the wollastonite samples come to be covered with newly formed apatite (HA) layers, and a continuous deposit of dense apatite took place over time. In SBF-treated wollastonite, coupled with wollastonite’s peaks, HA’s characteristic peak at 2θ position 31.79°, and this observation are in good agreement with the previous study.
Farah ′Atiqah et al. [76]	The pseudo-wollastonite peaks at 2θ angles 27, 33, 36.6, 45.7, and 62.9° were decreased from day 1 to day 7. The Ag peak, on the other hand, increased with increasing immersion time. After seven days of immersion, the presence of a significant peak means that the HA peak has begun to form. The HA peak was visible after 14 days of immersion, at a peak angle of 31–33°.
Roslinda et al. [77]	These findings suggest that the crystallinity of β-wollastonite decreased as the soaking time increased. The amorphous calcium phosphate (ACP) layer was discovered on day 3, and almost completely covered the wollastonite’s surface. An ACP structure is characterized by a broad peak centered between 30.0 and 35.0 degrees. The diminishing peak of β-wollastonite at 30.0 degrees verified this situation. The converted, unstable ACP structure began to form hydroxyapatite peaks (ICDD no. 72-1243), and no β-wollastonite or ACP structure peak was observed. The HA peak was discovered after just 21 days of soaking. The XRD pattern demonstrates that β-wollastonite transitions from a crystalline to an amorphous structure during the soaking phase.
Ismail et al. [79]	When β-wollastonite samples are immersed in SBF solution, apatite forms on their surfaces; after immersing β-wollastonite samples in SBF solution, two types of calcium phosphate groups were formed: amorphous calcium phosphate (ACP)—which is unstable—and calcium-deficient hydroxyapatite (CDHA).
Naghizadeh et al. [84]	The SEM micrographs confirmed that all silicate-based bioactive glass-ceramic (R-SBgC) scaffolds induced microsized apatite formation after SBF immersion. After 14 and 21 days, there was no substantial improvement in the pure polycaprolactone (PCL) scaffold.
Nayak et al. [85]	These crystalline phases are not present in specimens after soaking with SBF for three days. After 3 days of soaking, the specimens revealed the presence of an amorphous glassy phase and several crystalline calcium-phosphate-based phases. The specimens contained carbonated hydroxyapatite and hydrated calcium phosphate phases. The presence of these two phases increased from 14 to 21 days.
Nayak et al. [86]	In BGC900, these crystalline phases were almost non-existent. This is due to crystalline phase dissolution in SBF. The BGC900 included carbonated hydroxyapatite (Ca_10_(PO_4_)_3_(CO_3_)_3_(OH)_2_ and hydrated calcium phosphate (CaHPO_4_(H_2_O)_2_) phases. SBF incubation seems to have less impact on the crystalline peak of glass-ceramics sintered at higher temperatures. When glass-ceramics are sintered at higher temperatures, the crystalline phases can be less soluble in SBF.

**Table 11 materials-14-05193-t011:** Summary of studies on the in vitro biocompatibility of wollastonite derived from RHA and RSA.

Ref.	Biocompatibility Study
Choudhary et al. [22]	Hemolysis assay on the wollastonite was performed using the ASTM 756-00 and ISO 10 993–51,992 standards; both standards state that a sample is considered non-hemolytic if the hemolytic index range is less than 2%, slightly hemolytic if the range is between 2% and 5%, and hemolytic if the range is more than 5%. The results of the hemolysis assay show that, after 1 day of incubation, wollastonite was found to be hemocompatible at all concentrations (62.5, 125, and 250 g/mL). Nevertheless, after 24 h of incubation, a tendency toward increased RBC destruction was observed in the case of wollastonite.
Hossain et al. [23]	The author said that the hemolysis assay was carried out in accordance with ASTM F 756-00. Both waste-derived specimens had strong blood compatibility, with hemolysis indexes of less than 2% for 5 mg/mL concentrations.
Palakurthy et al. [73]	MTT assay of MG63 cells cultured with ceramic samples at different concentrations (50–1000 μg/mL) was used to examine cytocompatibility. A cell culture study showed that NCS (wollastonite from RHA and eggshells) ceramic particles are biocompatible and can proliferate in cells.
Palakurthy et al. [74]	The MTT assay was used to measure cytocompatibility, and the findings show that the synthesized wollastonite ceramic has no biological cell toxicity.
Farah ‘Atiqah et al. [76]	In this analysis, using 100% leachate with the medium leached for 1 and 3 days from each scaffold resulted in the death of all cells within 24 h of incubation using the PrestoBlue cell viability assay. A significant observation was that even though the cytotoxicity test showed a low growth rate, suggesting that the composite was toxic at the cellular level, there was a substantial increase in cell proliferation after 72 h.
Roslinda et al. [77]	β-wollastonite demonstrated biocompatibility with cells in the cell viability test.
Saravanan et al. [78]	The MTT findings demonstrated that the m-WS particles were nontoxic when revealed to mMSCs at different time points.
Alshatwi et al. [80]	The biocompatibility of the bSNPs was assessed using an MTT assay. For 24 and 48 h, the hLFCs were exposed to varying concentrations of bSNPs (25, 50, 100, 200, and 400 μg/mL). An in vitro model was used to conduct preliminary studies on the biocompatibility of rice-husk-derived, highly pure bSNPs with hLFCs. The cell viability results showed that the bSNPs did not affect the hLFCs. Lin et al. found that synthetic silica particles with diameters between 15 and 46 nm had important cytotoxic effects at low concentrations (50 μg/mL) [122]. The bSNPs permitted more than 85% cell viability in this study, even at concentrations as high as 400 g/mL, indicating that the bSNPs are biocompatible.
Athinarayanan et al. [82]	MTT assay was used to determine the biocompatibility of prepared bSNPs. According to the cell viability outcomes, bSNPs have great biocompatibility with hMScs.
Azeena et al. [83]	The particles’ cytocompatibility was investigated via MTT assay using murine mesenchymal stem cells at various concentrations and time intervals. The substance was found to be cyto-friendly up to a concentration of 0.05 mg/mL.

## Data Availability

No new data were created or analyzed in this study. Data sharing is not applicable to this article.

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
