# Peer review of "Bioactivity and Biocompatibility Properties of Sustainable Wollastonite Bioceramics from Rice Husk Ash/Rice Straw Ash: A Review"

_materials, 2021, doi:10.3390/ma14185193_

Round 1

Reviewer 1 Report

The article, by H.Ismail and H.Mohamad, presented for consideration is a review concerning to bioactivity and biocompatibility properties of wollastonite bioceramics obtained from ashes of rice husk and rice straw.

Remarks:

  1. Section Introduction needs to be revised.For example, in Line 38 authors say:

 “It is used as a biomaterial, biomedical implant, and surgery applications because agricultural waste adds considerable value.”  Which material? What is used as a biomaterial or biomedical implant? The paragraph, which includes line 43 to line 50 repeats what has been said before.

  1. Section 4 provides a detailed overview of the materials containing silica, used for the synthesis of wollastonite as well as the synthesis technologies. I would recommend that a comparative analysis be made at the end of Section 4 to point out the advantages and disadvantages of the feedstocks and especially of the synthesis methods used.
  2. Subsection 4.2.5 refers to mechanochemical synthesis rather than simple digestion of the raw materials. Mechanochemical synthesis is a variant of solid-phase synthesis, but it has indisputable advantages in comparison to it. This should be emphasized in the analysis of Section 4 proposed above.
  3. Table 6: The table presented in this form is incomprehensible. Maybe it is better to use the sign “+” instead of the symbol “/”.

The work is original and very interesting. It is an indisputable contribution in the field of the use of rice husks for the production of high-tech materials with applicability in various fields, incl. in the medicine. The work is comprehensive, very well organized and it is based on 104 current literature sources. After a minor revision, it is suitable for publication in journal Materials.

Author Response

Dear reviewer,

Thank you.

Reviewer 2 Report

In this work, a comprehensive review was conduced in the field of wollastonite (calcium silicate) production from silica-rich agricultural residues such as rice husk and rice straw ashes, and their biocompatibilities and bioactivities as a potential sources for dental and bone substitution. The quality of this work is high since it correctly addresses the current issue in managing such agricultural wastes. However, the reviewer think that the quality of the review can be improved if the following comments are also be considered.

  1. Introduction section includes discussion on valuable productions from agricultural wastes, particularly rice husk (RH) and rice straw (RS). However, it does not cover one of the common usage of RH and RS, which is biogenic silica production [1]. It is highly recommended to also include generation of biogenic silica, as a value-added product from thermochemical conversion of RH and RS.
  2. It seems that Fig. 1 is not needed since it does not include further scientific information. Therefore, it is recommended to remove this figure.
  3. In lines 95 and 99, it is better to change “1st generation: Inert” and “2nd generation: Bioactive and bioresorbable” to “1st generation: Inert bioceramics” and “2nd generation: Bioactive and bioresorbable bioceramics”.
  4. Font of the text should be uniform. Please use only one font type and size in the entire manuscript.
  5. Quality of the presented phase diagram (Fig. 4) is so poor. It should be replaced with a high quality image.
  6. In line 153, the number of mentioned figure is not correct.
  7. Average ash content of rice husk and rice straw are both in the range of 16 wt.% in dry based [1]. It is recommended to report a reliable value for all the parameters in Table 3.
  8. In line 216, the reported world paddy rice production seems to be lower than the commonly reported values by the Food and Agriculture Organization of the United Nations (FAO) shown in a figure in [1]. It is better to cite a reliable source with correct actual production value.
  9. There is no need to report already published figure (Fig. 5) since it does not have a significant input to the discussion. A similar figure was also published in [1], reporting the FAO’s data.
  10. In Table 5, the raw material of the reported investigation for Saravanan et al. [60 in the text], was rice husk ash and limestone. Please check the other references and correct the reported information.
  11. Polymerization networks presented in lines 331-332 have very low quality. Please redraw them and provide a high-quality images.
  12. It is better to replace “/” with “Yes” or “No” in Table 6 to present the bioactivity and biocompatibility tests by the reported research works. Furthermore, details of these two type of tests should also be included in the manuscript as it is one of the major aim of the present article.
  13. The quality of Fig. 8 is really poor. It should be redrawn or high-quality images should be presented instead of the current ones. The same comment is also valid for Fig. 9.
  14. In section 6, as a future perspective, it is correctly mentioned that “Hopefully, the commercialization of wollastonite products based on RHA/RSA would occur.” It is also recommended to highlight that most of the attempt for the production of biogenic silica were conducted in lab scale [1], and a few numbers of studied such as the investigation by Schliermann et al. [2] was performed in large scales. The number of such a big scale investigation should be increased to secure sustainable silica and Wollastonite production from silica-rich agricultural residues.

[1] https://www.mdpi.com/2076-3417/9/6/1083

[2] Schliermann, T.; Hartmann, I.; Beidaghy Dizaji, H.; Zeng, T.; Schneider, D.; Wassersleben, S.; Enke, D.; Jobst, T.; Lange, A.; Roelofs, F.; et al. High quality biogenic silica from combined energetic and material utilization of agricultural residues. In Proceedings of the 7th International Symposium of Energy from Biomass and Waste, Venice, Italy, 15–18 October 2018; ISBN 978-8-86-265013-7.

Author Response

Dear reviewer,

Thank you.

Reviewer 3 Report

The paper is focused on obtaining crystalline wollastonite from biodegradable raw materials of rice hulls. The study is very relevant, due to the large number of raw materials in the processing of rice. The work deserves to be published in “Materials”, but with additions.

  1. The reviewed paper is a topic of highly actual and the list of literature consisting of 104 sources is still deserves to be expanded with additional review of the current literature sources of the last two years (2020–2021).
  2. Do the authors have data confirming the concentration of silicon in the composition of rice husks, in particular, for the samples presented in Figure 6. Are TG data available for these samples?
  3. The description of the process of wollastonite synthesis is of interest.
  4. This review deserves to be extended to mention the consolidation techniques of wollastonite using a variety of methods: cold pressure and sintering, HIP, microwave sintering, spark plasma sintering, and other methods of compacting and consolidation of the original powder raw materials. Please familiarize yourself with work in obtaining powders and ceramics [10.1016/j.ceramint.2021.04.258; 10.1016/j.powtec.2020.04.040; 10.1039/C6RA04956G].
  5. Concerning the simulated body fluid in-vitro method, the study of the phosphate formation on the samples in dynamics (kinetics of formation, sampling intervals, a study of the apatite layer) is of great interest.
  6. Is the graph shown in Figure 8 a borrowing? The source of borrowing should be indicated in the caption.
  7. Of the minor comments concerning the aesthetic component of the material presented in the article, to improve readability it may be recommended:

- in Figure 4, present the phase diagram in better resolution and be sure to indicate the source of its borrowing in the figure caption.

- for the content of the column "bioactivity study" in Table 9 to improve the perception of the text to change the alignment (on the edge).

Author Response

Dear reviewer,

Thank you.

Round 2

Reviewer 3 Report

Dear authors, 

thank you for your corrections and additions to the paper. In the updated form presented, the paper may be published in the journal "Materials". I wish you all the best in your continued work!